# Sister kinetochore splitting and precocious disintegration of bivalents could explain the maternal age effect

Agata P Zielinska[1], Zuzana Holubcova[1], Martyn Blayney[2], Kay Elder[2], Melina Schuh[1,3]*

[1]Medical Research Council Laboratory of Molecular Biology, Cambridge, United Kingdom; [2]Bourn Hall Clinic, Cambridge, United Kingdom; [3]Max Planck Institute for Biophysical Chemistry, Goettingen, Germany

**Abstract** Aneuploidy in human eggs is the leading cause of pregnancy loss and Down's syndrome. Aneuploid eggs result from chromosome segregation errors when an egg develops from a progenitor cell, called an oocyte. The mechanisms that lead to an increase in aneuploidy with advanced maternal age are largely unclear. Here, we show that many sister kinetochores in human oocytes are separated and do not behave as a single functional unit during the first meiotic division. Having separated sister kinetochores allowed bivalents to rotate by 90 degrees on the spindle and increased the risk of merotelic kinetochore-microtubule attachments. Advanced maternal age led to an increase in sister kinetochore separation, rotated bivalents and merotelic attachments. Chromosome arm cohesion was weakened, and the fraction of bivalents that precociously dissociated into univalents was increased. Together, our data reveal multiple age-related changes in chromosome architecture that could explain why oocyte aneuploidy increases with advanced maternal age.

*For correspondence: melina.schuh@mpibpc.mpg.de

**Competing interests:** The author declares that no competing interests exist.

## Introduction

Once during every menstrual cycle, an oocyte segregates half of its chromosomes into a small cell, the polar body, to prepare for fertilization (*Petronczki et al., 2003*, *Clift and Schuh, 2013*). Segregating the chromosomes accurately is vital for the development of healthy human embryos upon fertilization. Surprisingly, oocytes segregate their chromosomes much more inaccurately than do human mitotic cells (*Nagaoka et al., 2012*, *Knouse et al., 2014*), human spermatocytes (*Pacchierotti et al., 2007*, *Templado et al., 2011*) or mouse oocytes (*Pacchierotti et al., 2007*, *Danylevska et al., 2014*, *Hassold and Hunt, 2001*). The accuracy of chromosome segregation drops even further with increasing age (*Nagaoka et al., 2012*, *Chiang et al., 2010*, *Fragouli et al., 2011*, *Kuliev et al., 2011*). This may be one of the reasons behind an increase in miscarriages, trisomic pregnancies and infertility with advanced maternal age.

Notably, eggs from young women are also often aneuploid (*Pacchierotti et al., 2007*, *Obradors et al., 2011*). Why chromosomes missegregate so frequently, even in oocytes from young women, is still poorly understood. We showed previously that human oocytes assemble a spindle by an unusually long and error-prone mechanism that is likely to sensitize human oocytes of all ages to abnormal kinetochore-microtubule attachments and chromosome segregation errors (*Holubcova et al., 2015*). However, it is still unclear why chromosome segregation errors in oocytes become more frequent as women get older.

Research in mouse oocytes suggests that changes in chromosome architecture contribute to this age-dependent increase in aneuploidy (*Lister et al., 2010*, *Chiang et al., 2011*, *Yun et al., 2014*,

**eLife digest** Older women are more likely to experience a miscarriage or give birth to a child who has a developmental disorder. This occurs because age increases the chances that a woman's egg cells will have the wrong number of chromosomes. If a sperm fertilizes an egg with too many or too few copies of a chromosome, the resulting embryo will have the wrong number of copies for many genes. Many of these embryos fail to develop and die, but some are born with developmental conditions like Down's syndrome and Turner syndrome.

New egg cells develop from immature egg cells that are present in a woman from birth. In an immature egg cell, chromosomes that came from the woman's father are paired up with the matching chromosomes from the woman's mother and the handle-like structures on each chromosome (called the kinetochores) are fused. Just before the immature egg cell divides, a molecular machine called 'the spindle' attaches to the chromosome handles. The spindle then separates these pairs of chromosomes such that each new cell receives only one copy of each chromosome. However, while it is known that this process sometimes goes wrong, it is not clear why mistakes happen more often in older women.

Now, Zielinska et al. used powerful microscopes to observe cell division in over 200 preserved or living immature egg cells donated by women between the ages of 23 and 46. First, the experiments examined over 1,000 chromosomes in preserved immature egg cells that were about to divide. This revealed that the chromosome handles that were supposed to be fused had often disconnected in women over 35 years old. Chromosome pairs without correctly fused handles were also prone to rotating during the division process, and sometimes the pairs simply fell apart too soon.

Further experiments with living immature egg cells then revealed that the spindle struggled to grip and separate the chromosomes correctly, possibly because the chromosome handles were not properly fused. These events increased the likelihood of a new egg cell receiving too many or too few chromosomes. Finally, Zielinska et al. found that immature egg cells lack a robust control mechanism that can detect when these problems occur.

Together these findings help to explain why miscarriages and chromosome abnormalities are more common in the children of older women. Research building on these findings may in the future help women in their late 30s and early 40s to increase their chances of having a family.

Sakakibara et al., 2015). The architecture of chromosomes differs in meiosis and mitosis. During the first meiotic division, the oocyte segregates entire chromosomes instead of sister chromatids as is the case in mitosis. To facilitate this, the chromosome pairs are linked with each other through meiotic recombination (Nagaoka et al., 2012, Kong et al., 2004, Baudat et al., 2013). The sister kinetochores of each chromosome become unified so that they function as a single kinetochore (Watanabe, 2012). This prevents the undesirable bi-orientation of sister kinetochores during meiosis I (Watanabe, 2012, Yokobayashi and Watanabe, 2005, Kim et al., 2015). The resulting unit of two linked chromosomes with two functional kinetochores is called a bivalent chromosome. The cohesion complex has a crucial role in maintaining the integrity of the bivalents: it holds the homologous chromosomes together and promotes the close association of the sister kinetochores (Petronczki et al., 2003, Watanabe, 2012).

Recent work in mouse oocytes suggests that the cohesin complex is gradually lost from oocyte chromosomes as mice get older (Lister et al., 2010, Tachibana-Konwalski et al., 2010, Chiang et al., 2010). This loss of cohesin has been suggested to lead to an increase in sister kinetochore distances and the precocious dissociation of bivalents into individual chromosomes (Yun et al., 2014, Sakakibara et al., 2015), often referred to as univalents in meiosis. Whether such a loss of cohesin is also relevant in human oocytes is still unclear (Garcia-Cruz et al., 2010, Tsutsumi et al., 2014, Handyside et al., 2012).

The aim of this study was to investigate whether age-related changes in chromosome architecture also contribute to the maternal age effect in human oocytes. To this end, we analyzed the organization of chromosomes and how they interact with the spindle in over 200 live and fixed human oocytes from donors aged between 23 and 46 years.

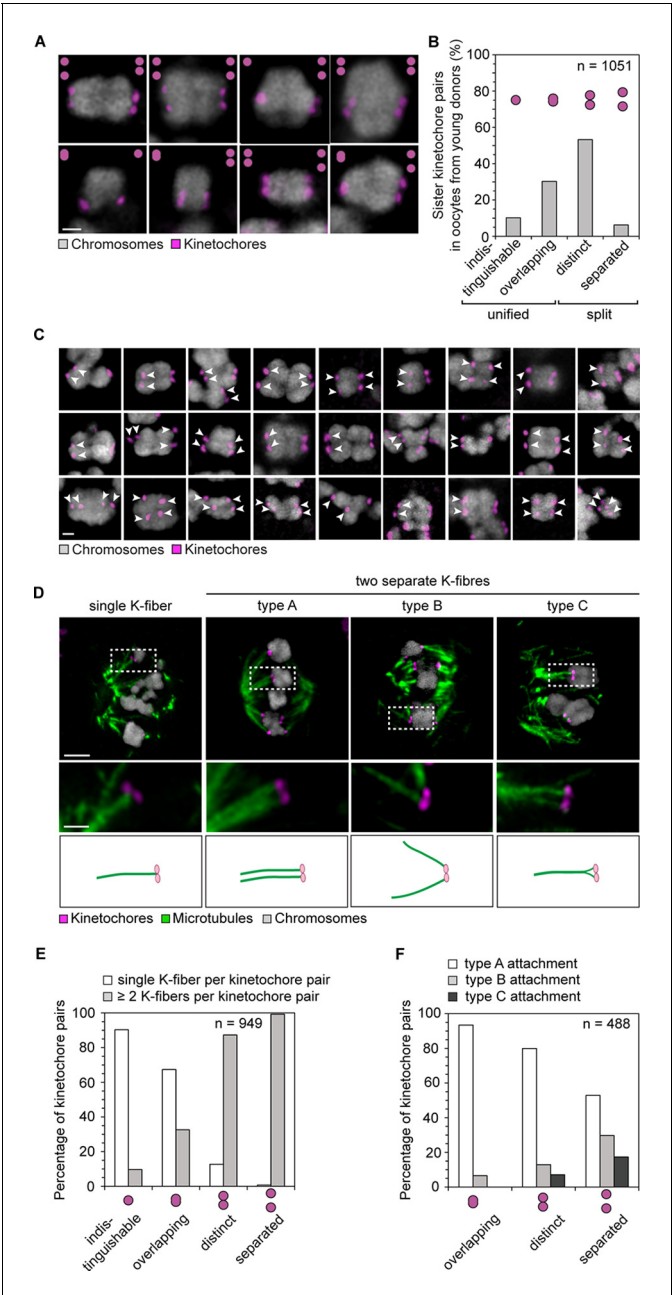

**Figure 1.** Most sister kinetochores are not unified and attach to two discrete k-fibers during meiosis I in human oocytes. (A) Maximum intensity z-projection immunofluorescence images of kinetochores and chromosomes in meiosis I oocytes from young donors (≤30 years old). Representative examples from 5 different oocytes. Scale bar represents 1 µm. (B) Categories of sister kinetochore configurations shown in (A) and their frequency in oocytes from young donors (≤30 years old). 1,051 kinetochore pairs from 23 oocytes were analyzed. (C) Examples of kinetochore configurations from 26 human oocytes across all age groups in which sister kinetochores are separated (marked with white arrows). Scale bar represents 1 µm. (D) Maximum intensity z-projection immunofluorescence images of microtubules and distinct sister kinetochore pairs in cold-treated meiotic spindles from human oocytes across all age groups. Drawings illustrate the different modes of microtubule attachments in the immunofluorescence images. Scale bar represents 5 µm in overview and 1 µm in insets. (E) Quantification of the number of individual k-fibers attaching to sister kinetochore pairs. 949 attachments from 25 cold-treated human oocytes from donors across all age groups were evaluated. (F) Frequency of the different modes of kinetochore-microtubule attachments shown in (D).

The following figure supplements are available for figure 1:

**Figure supplement 1.** The majority of sister kinetochores are split during meiosis I in human oocytes.

*Figure 1 continued on next page*

*Figure 1 continued*

**Figure supplement 2.** Sister kinetochore separation is already evident in the initial stages of spindle assembly.

**Figure supplement 3.** Sister kinetochores can interact with k-fibers that originate from different regions of the meiotic spindle.

**Figure supplement 4.** Individual kinetochores can attach to multiple k-fibers during meiosis I in human oocytes.

## Results

We first investigated the chromosome organization in oocytes from young women (≤30 years) by recording high resolution z-stacks (300 nm in x and y, 650 nm in z) of 1,051 chromosomes and their kinetochores in 23 oocytes. Unexpectedly, 60% of sister kinetochores appeared as two discrete spots when observed by light microscopy (*Figure 1A,B*). This is in contrast to oocytes from young mice, whose sister kinetochores generally appear as a single spot (*Chiang et al., 2010*, *Lister et al., 2010*, *Shomper et al., 2014*). The sister kinetochores were spaced up to 1.7 μm apart (*Figure 2C*) and sometimes separated by prominent gaps (*Figure 1A,B*). Such bivalents with separated kinetochores were a frequent phenomenon, observed throughout the entire cohort of oocyte donors (*Figure 1C*; *Figure 1—figure supplement 1,2*), including very young donors (*Video 1*).

We then analyzed how the split kinetochores interact with the spindle. Around 90% of split kinetochores were attached to two separate kinetochore fibers (*Figure 1D,E*). The two kinetochore fibers did not always run in parallel, but often linked the sister kinetochores to distant positions at the spindle poles (*Figure 1D,F*). Thus, consistent with the high degree of separation, the spindle recognizes most split sister kinetochores in human oocytes as independent units (*Figure 1—figure supplement 3*,*4*).

The fraction of split kinetochores in human oocytes increased with maternal age. Around 75% of all kinetochores were split in oocytes from women between 30 and 35. This increased to 87% in women over 35 (*Figure 2A–E*; *Videos 1*, *2*). The fraction of sister kinetochores that was separated from each other by a prominent gap also increased dramatically with advanced maternal age (*Figure 2D*). Sister kinetochores also sometimes become distinguishable in oocytes from very old mice (*Chiang et al., 2010*, *Lister et al., 2010*). But in contrast to human oocytes, their sister kinetochores are not separated by a prominent gap. Consistent with the remarkably large distances between sister kinetochores during meiosis I (*Figure 2C*, *Figure 2—figure supplement 1*) and meiosis II (*Duncan et al., 2012*) in human oocytes, sister kinetochores were also separated by very large gaps during meiosis II (*Figure 2—figure supplement 2*; *Videos 3*, *4*).

Together, these data show that the majority of bivalent chromosomes in human oocytes feature three or four functionally distinct kinetochores. Each of these kinetochores is able to establish its own kinetochore fiber. Thus, human oocytes question the paradigm that sister kinetochores are unified during meiosis I.

We then went on to analyze whether the split nature of sister kinetochores in human oocytes could affect chromosome segregation. First, we assessed if having three or four functionally distinct kinetochores per bivalent increased the probability of abnormal kinetochore-microtubule attachments. During meiosis I, the sister kinetochores of each chromosome should be attached to a single spindle pole only (amphitelic attachment). In this way, they can be segregated accurately after anaphase onset. If a pair of sister kinetochores is attached to opposite spindle poles instead (merotelic attachment), the chromosome will be pulled from both sides of the spindle upon anaphase onset. Such pulling will cause a chromosome to lag behind during anaphase (*Cimini et al., 2001*, *Lane et al., 2012*). This can lead to aneuploidy when the ingressing cytokinetic furrow partitions chromosomes inaccurately between the egg and the polar body. To assess how the bivalent chromosomes were attached to the spindle, we briefly placed the oocytes on ice before fixation. This selectively preserves the more stable microtubules that interact with kinetochores (*Zhai et al., 1995*). We then stained the oocytes for microtubules, chromosomes and kinetochores.

Separated sister kinetochores were more likely to be merotelically attached to the spindle than unified sister kinetochores. Interestingly, we observed that the degree of separation correlated with the probability of a pair being merotelically attached: the fraction of merotelic attachments increased from 12% in unified kinetochore pairs to 17% when the sister kinetochores were distinct

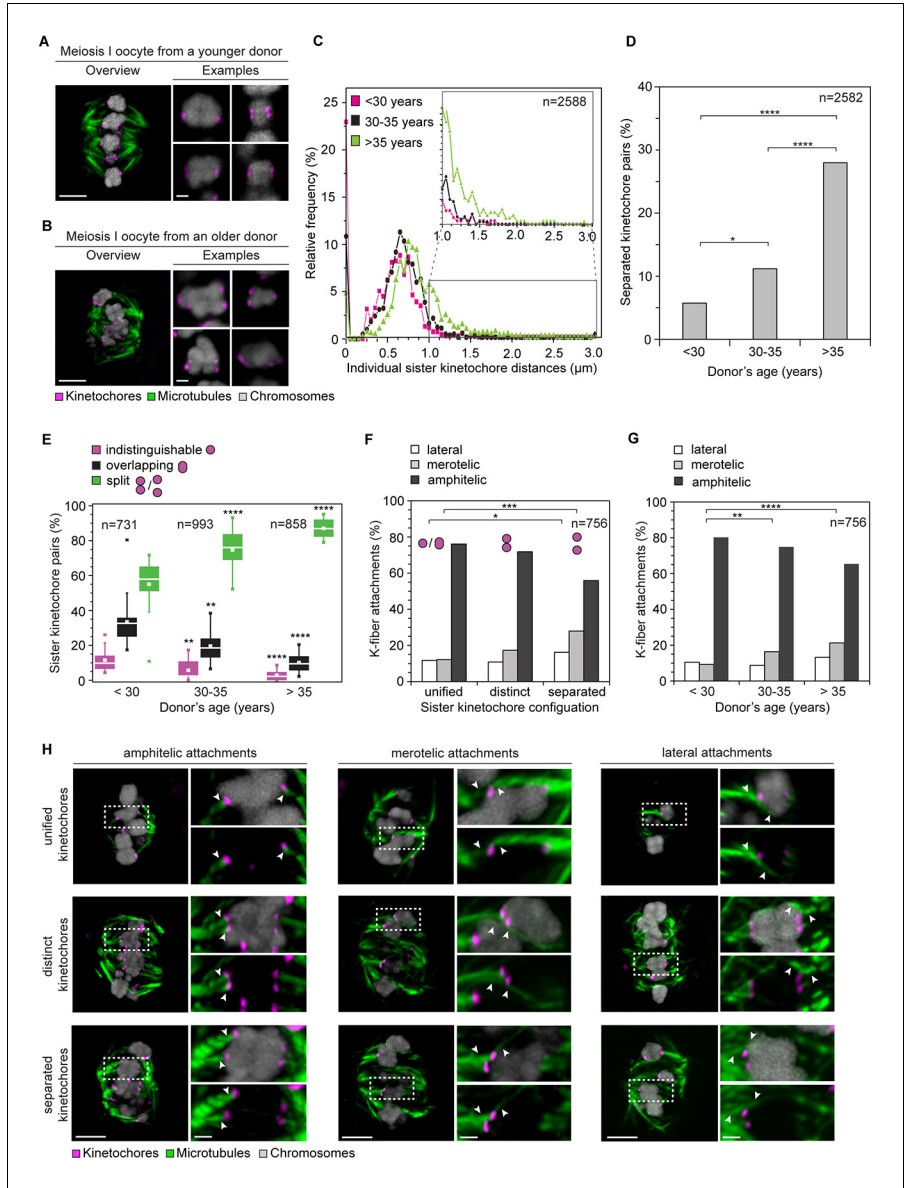

**Figure 2.** Age-dependent increase in sister kinetochore separation correlates with abnormal kinetochore-microtubule attachments. (A, B) Overview of a cold-treated meiotic spindle and examples of its bivalents in an oocyte from (A) a young (24 years old) and (B) an older (40 years old) donor. Scale bars represent 5 μm in overviews and 1 μm in examples. (C) Relative frequency of individual sister kinetochore pair distances across different age groups. 2,588 measurements were obtained from 57 oocytes. Distances between unresolvable sister kinetochore configurations of the indistinguishable and overlapping categories (as shown in (1B)) are presented as 0 μm. (D) Age-related increase in the incidence of separated sister kinetochores in 58 human oocytes. ****$p \leq 0.0001$. (E) Occurrence of kinetochore configurations defined in (1B) across different age groups. *$p \leq 0.05$, **$p \leq 0.01$, ***$p \leq 0.001$, ****$p \leq 0.0001$. Significance analyses were performed for changes relative to <30 years old age group. (F) Quantification of the proportion of microtubule attachments (shown in H) in each category of sister kinetochore configurations. *$p \leq 0.05$, **$p \leq 0.01$, ***$p \leq 0.001$ (Fisher's exact test). Significance analyses were performed for changes in the frequency of merotelic (***) and lateral (*) attachment modes relative to the unified kinetochore group. (G) The proportion of kinetochore-microtubule attachments (shown in (H)) in cold-treated spindles of oocytes from each age group. N=25 oocytes. *$p \leq 0.05$, **$p \leq 0.01$, ***$p \leq 0.001$, ****$p \leq 0.0001$. Significance analyses were performed for changes in the frequency of merotelic attachment mode relative to <30 years old age group. (H) Representative maximum intensity z-projections of immunofluorescence images of cold-treated spindles showing different microtubule attachment types across the three categories of sister kinetochore configurations defined in (1B). Arrowheads indicate kinetochore pairs with the specified microtubule attachment type. Scale bar represents 5 μm in overview, 1 μm in insets.

The following figure supplements are available for figure 2:

**Figure supplement 1.** Mean sister kinetochore distance in meiosis I increases with donor's age in a linear manner.

*Figure 2 continued on next page*

*Figure 2 continued*

**Figure supplement 2.** Age-dependent increase in separation of sister kinetochores in human oocytes is conserved from meiosis I to meiosis II.

**Figure supplement 3.** Frequency of merotelic attachments are correlated with kinetochore configuration and donor's age.

but still linked, and up to 28% in completely separated kinetochore pairs (*Figure 2F,H*). This could potentially be due to improved accessibility from the opposite spindle pole, when two tightly joined sister kinetochores are compared with two separated sister kinetochores.

The number of merotelic kinetochore-microtubule attachments also increased with maternal age, from around 7% in women under 30 to around 21% in women over 35 (*Figure 2G*). This is consistent with the increased fraction of separated sister kinetochores with advanced maternal age. Separated kinetochores were more likely to be merotelically attached in oocytes from older women than in oocytes from young women (*Figure 2—figure supplement 3*). This could be explained by the increase in the degree of separation between sister kinetochores as women get older (*Figure 2C*), as well as additional age-related defects that could contribute to aberrant attachments.

The remarkably large distances between some sister kinetochores raised an unexpected problem: a stable arrangement of bivalent chromosomes on the spindle could theoretically also be achieved if the split sister kinetochores are facing towards opposite poles instead of the same spindle pole (*Figure 3A* scheme, middle row; fully-rotated). Such a chromosome would be rotated on the spindle by 90 degrees and could not segregate accurately during anaphase.

To our surprise, a large fraction of oocytes had one or more rotated bivalent chromosomes, in which the sister kinetochores of both pairs faced opposite spindle poles (*Figure 3A–C*, middle row; *Figure 3D*; *Figure 3—figure supplement 1*; *Videos 5, 6*). Given that the orientation of sister kinetochores in these rotated bivalents is inverted, we called these bivalents 'inverted bivalents'. We also often observed half-inverted bivalents, in which only one pair of the sister kinetochores was oriented towards opposite spindle poles (*Figure 3A–C* bottom row, half-inverted; *Figure 3—figure supplement 1D*, right panel), while the other pair was correctly oriented, with both sister kinetochores facing a single spindle pole (*Figure 3D*). Bivalent rotation was also evident from the distribution of Shugoshin-1, which marked the pairs of sister kinetochores (*Figure 3C*, *Figure 3—figure supplement 1*; *Videos 5,6*). Interestingly, Shugoshin-1 associated with a wider domain than in meiosis I mouse oocytes (*Lee et al., 2008*). It will be interesting to investigate how meiotic cohesion is achieved in human oocytes and whether the unexpected kinetochore configurations that we observed could be linked to species-specific differences in this process.

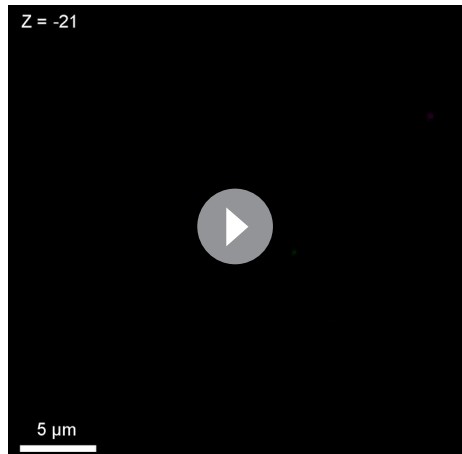

**Video 1.** A projection through confocal sections of a meiosis I spindle in an oocyte from a young donor (24 years old) stained for microtubules (green), chromosomes (grey) and kinetochores (magenta).

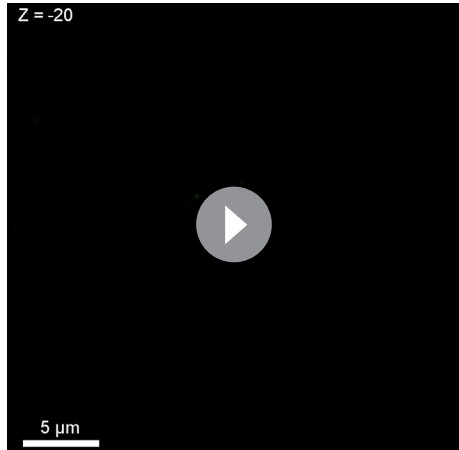

**Video 2.** A projection through confocal sections of a meiosis I spindle in an oocyte from an older donor (40 years old) stained for microtubules (green), chromosomes (grey) and kinetochores (magenta).

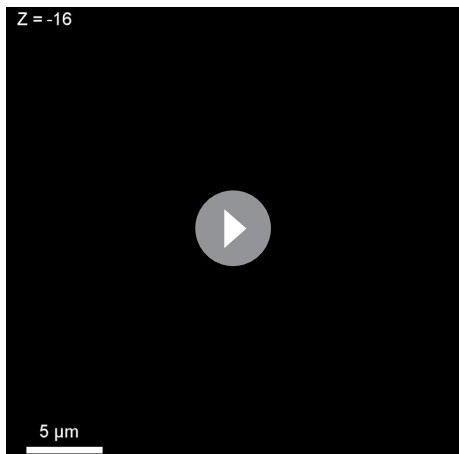

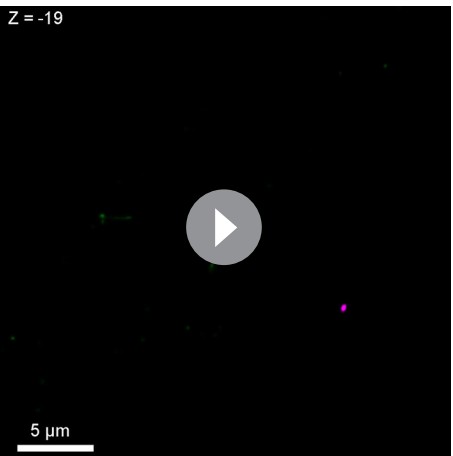

**Video 3.** A projection through confocal sections of a meiosis II spindle in an oocyte from a young donor (23 years old) stained for microtubules (green), chromosomes (grey) and kinetochores (magenta).

**Video 4.** A projection through confocal sections of a meiosis II spindle in an oocyte from an older donor (40 years old) stained for microtubules (green), chromosomes (grey) and kinetochores (magenta).

In total, more than 2% of all bivalents were half or fully inverted (*Figure 3E*). Importantly, kinetochores of inverted bivalents were linked to the spindle through cold-resistant and thus stable microtubules (*Figure 3—figure supplement 2A,C*). This indicates that the altered orientation of kinetochores in inverted bivalents does not completely compromise their ability to capture microtubules. Moreover, the incidence of inverted bivalents increased with maternal age, from 12.5% in oocytes from women under 30 to 40% in oocytes from women over 35 years old (*Figure 3F*; *Figure 3—figure supplement 1C*). This is consistent with the increased separation of sister kinetochores with maternal age, which is expected to promote the rotation (*Figure 3—figure supplement 2B,D*).

Having four instead of only two independent microtubule attachment sites also allowed bivalents to be twisted along their axis (*Figure 4A*). In around 20% of bivalents with split sister kinetochores, the chromosomes were twisted, as judged by the observation that their sister kinetochore pairs were oriented perpendicular instead of parallel to each other (*Figure 4B*). Such twisting is likely to exert additional forces on arm cohesion, which is possibly already weakened in oocytes at the time of meiotic resumption (*Duncan et al., 2012*, *Chiang et al., 2010*). Arm cohesion was indeed reduced in a large fraction of chromosome bivalents (*Figure 5A,C,D*). In total, 10% of all bivalents were weakly associated as evident from a gap in the bivalents' center. In some cases, the chromosomes were separated by a gap of a few 100 nm only. In other cases, gaps of up to 3.2 μm between two chromosomes were visible (*Figure 5A*). Despite the gap, the chromosome pairs were under tension, with each pair of sister kinetochores facing the correct spindle pole. This suggests that chromosomes with reduced arm cohesion still behave as functional bivalents.

Insufficient arm cohesion was also evident from the complete disintegration of 1.2% of bivalents into univalents (*Figure 5B–F*). 95% of these univalents aligned in the center of the spindle, with their sister kinetochores facing opposite spindle poles, suggesting that they could segregate equally into two sister chromatids upon anaphase onset (*Figure 5E*). The fraction of bivalents affected by loss of arm cohesion increased significantly with maternal age. In women under 30, 5% of all chromosome pairs were separated by gaps or completely disintegrated into univalents. This increased to 11% of chromosome pairs in women over 35 years old (*Figure 5D*). A recent study also reported the presence of univalents in oocytes from older women (*Sakakibara et al., 2015*). Our data suggests that the premature dissociation of bivalents into univalents also contributes to aneuploidy in oocytes from young women (*Figure 5F*). It also provides further evidence from a larger number of oocytes that an increase in univalents is a likely contributor to the maternal age effect, because more than 40% of oocytes from women over 35 featured univalents (*Figure 5F*).

Having more than two functional kinetochores on each bivalent chromosome is expected to hinder the correct attachment of chromosomes to the spindle. Live imaging in human oocytes showed

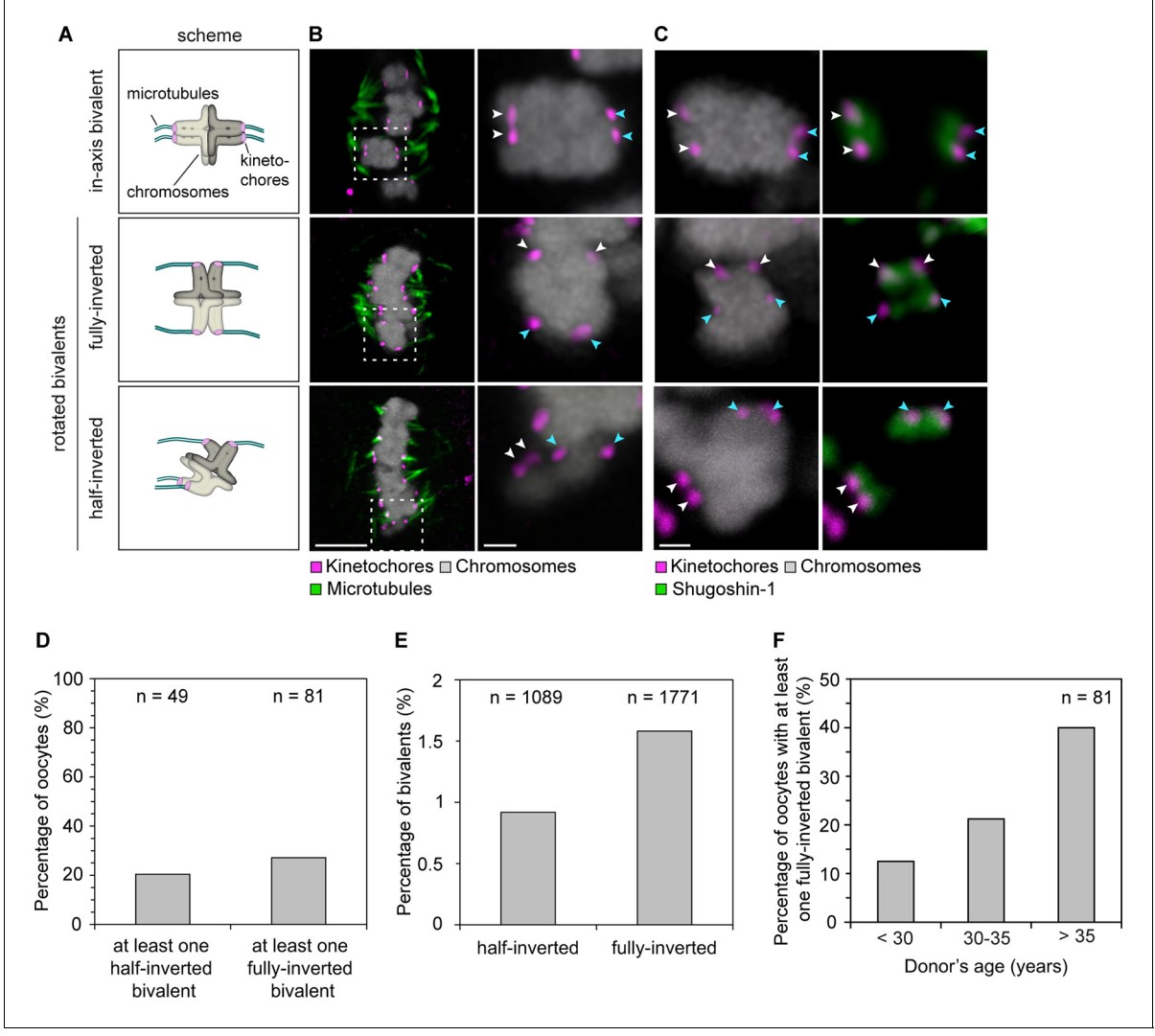

**Figure 3.** Sister kinetochore separation allows bivalents to rotate and twist on the meiotic spindle. (A) Schematic representation of the possible orientations of bivalents on the meiosis I spindle. (B) Representative images of the different orientations that bivalents can adopt relative to the axis of the metaphase I spindle. Arrows of the same colour highlight sister kinetochores. Scale bar represents 5 μm, 1 μm in insets. (C) Representative images of Shugoshin-1 staining in bivalents rotated relative to the axis of the metaphase I spindle. Arrows of the same colour highlight sister kinetochores. Scale bar represents 1 μm. (D) Occurrence of one or more inverted bivalents in fully assembled metaphase I spindles. Half-inverted bivalents were scored only in oocytes subjected to cold treatment which selectively preserves kinetochore fibers and hence allows for detection of bioriented kinetochore pairs. Fully-inverted bivalents were scored both in cold-treated and non-treated meiosis I spindles. (E) Proportion of bivalents that are fully- or half-inverted on late metaphase I spindles. (F) Occurrence of fully inverted bivalents in oocytes from donors across all age groups.

The following figure supplements are available for figure 3:

**Figure supplement 1.** Shugoshin-1 is a reliable marker of sister kinetochores in meiosis I human oocytes and can be used to identify rotated bivalents.

**Figure supplement 2.** Despite their non-conventional geometry, rotated bivalents can form stable k-fiber attachments.

that around 16.3 ± 2.3 hr were needed to align the chromosomes in the center of the spindle upon nuclear envelope breakdown (*Holubcova et al., 2015*). The period of chromosome congression and alignment coincided in time with spindle instability and reorganization (*Figure 6—figure supplement 1*). It is conceivable that the reorganization of the spindle is required to attach the bivalent chromosomes correctly to the spindle, consistent with a previous study in mouse oocytes (*Kitajima et al., 2011*).

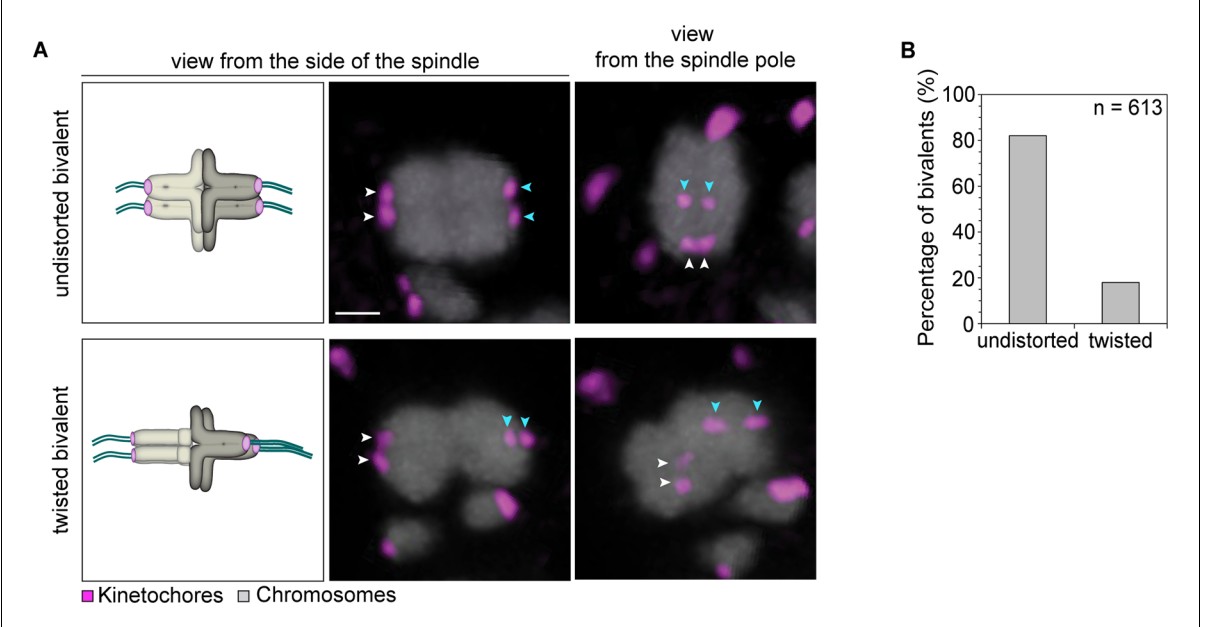

**Figure 4.** Kinetochore separation allows twisting of bivalents. (**A**) Representative three-dimensional images and schematic illustrations of undistorted and twisted bivalent geometries. Arrows of the same colour highlight sister kinetochores. Scale bar represents 1 µm. (**B**) Geometric categories of bivalents displaying a split sister kinetochore configuration (defined in (**1B**)). Undistorted and twisted geometries were defined as bivalents with kinetochore pairs that are parallel and perpendicular to each other, respectively.

Difficulties in correctly attaching bivalents to the spindle would only lead to aneuploidy if oocytes are allowed to progress into anaphase in the presence of abnormal kinetochore-microtubule attachments. Thus, a decisive question in understanding how aneuploidy arises is whether human oocytes have effective control mechanisms that monitor progression into anaphase. In mitosis, the spindle assembly checkpoint delays progression into anaphase until all chromosomes are correctly attached to the spindle (*Musacchio and Salmon, 2007*). Whether human oocytes have similar mechanisms is unclear. Live imaging revealed that all oocytes without chromosome alignment defects progressed into anaphase. Similarly, all oocytes with mild alignment defects and 92% of oocytes with severe misalignment progressed into anaphase (*Figure 6A,B*; *Videos 7*, *8*). Thus, the presence of chromosome alignment defects did not preclude progression into anaphase. In addition, the onset of anaphase was not severely delayed when misaligned or lagging chromosomes were present (*Figure 6C,D*). Less prominent delays might however be masked by some variability in anaphase timing between oocytes. Collectively, these results suggest that human oocytes progress into anaphase with normal efficiency and without major delays if chromosomes are misaligned.

## Discussion

Our findings suggest possible answers to the two key questions related to human oocyte meiosis. Firstly, why is chromosome segregation in human oocytes in general more error-prone than in mitosis, even in oocytes from young women? Our previous work established that the spindles in human oocytes form over a lengthy and error-prone process, during which the spindle undergoes extensive reorganization. Chromosomes frequently remain merotellically attached to the spindle close to anaphase onset. These merotelic attachments could be favoured by the high instability and morphology of the spindle in human oocytes (*Holubcova et al., 2015*). The data presented in this study suggest that merotelic attachments are also promoted by the split nature of sister kinetochores in human oocytes.

It is conceivable that the specialized architecture of bivalents in human oocytes and spindle reorganization are directly linked: the fact that the four sister kinetochores of a bivalent chromosome can interact with microtubules independently may make it more challenging for the oocyte to

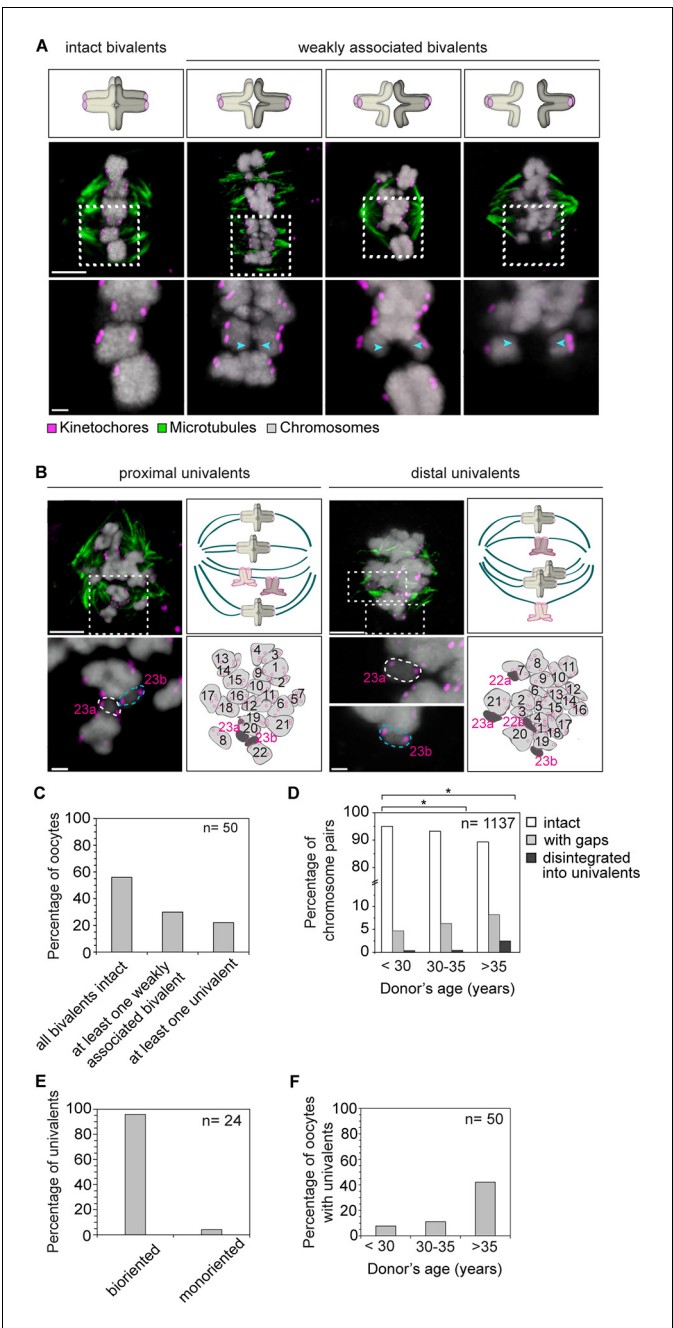

**Figure 5.** Precocious dissociation of bivalents into univalent contributes to aneuploidy. (A) Illustrations and representative images of homologs showing gaps between weakly-associated bivalents (indicated by arrowheads). Scale bar represents 5 µm in overviews and 1 µm in insets. (B) Representative immunofluorescence images and illustrations showing complete disintegration of bivalents into two proximally or distally positioned univalents. Lower panel insets and cartoons show univalents (dark grey, numbered in magenta) and bivalents (light grey, numbered in black) that were determined by manual tracing and identification of homologs within the spindle. Scale bar represents 5 µm in overviews and 1µm in insets. (C) Quantification of weakened chromosome arm cohesion and bivalent disintegration in 50 human oocytes. (D) Proportion of intact, weakly-associated (with gaps) or disintegrated bivalents amongst 1,137 chromosomes across different age groups. * $p \leq 0.05$ (Fisher's exact test). Significance analyses were performed for changes relative to <30 years old age group. (E) Proportion of univalents that achieved bi-orientation on the meiosis I spindle. (F) Proportion of human oocytes that contained at least one univalent pair across different age groups.

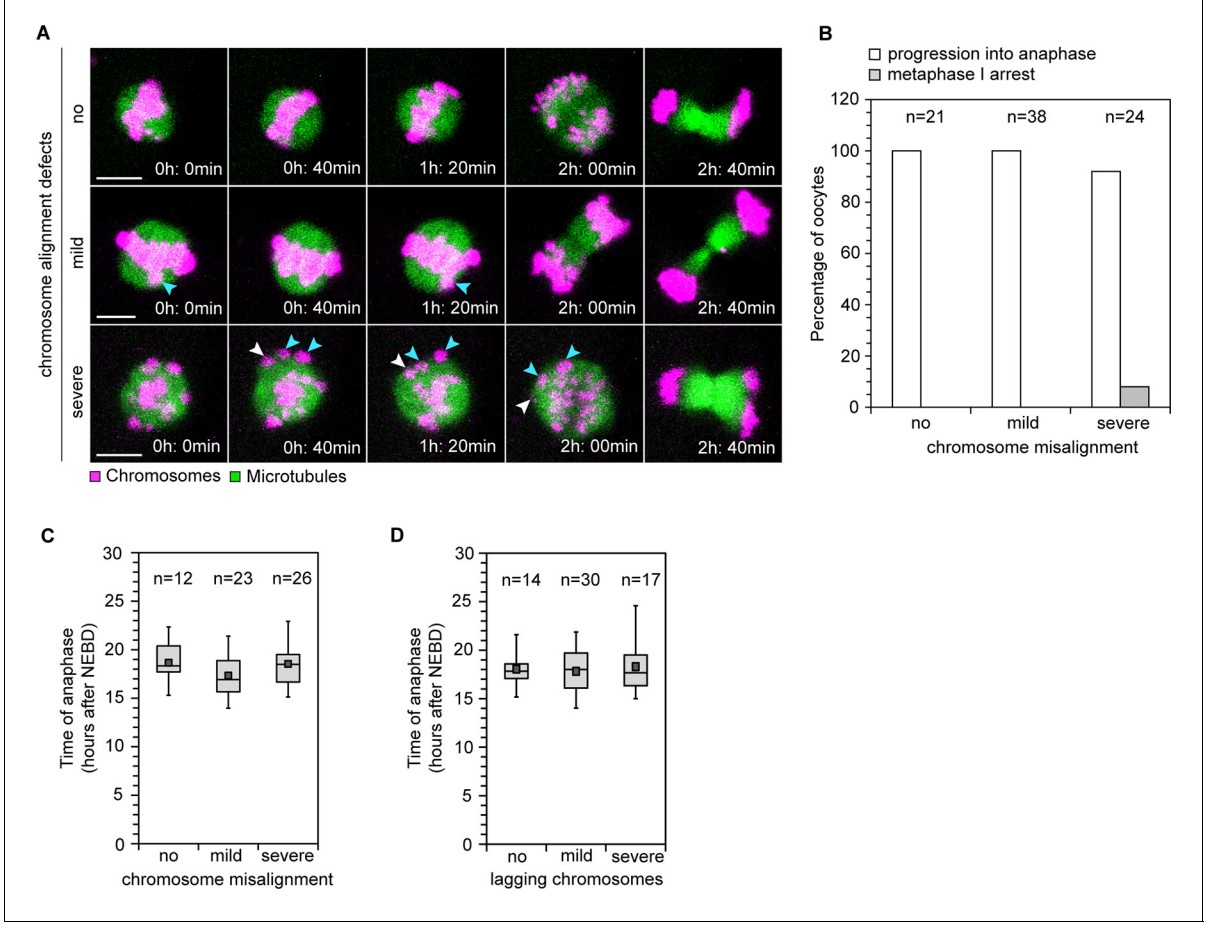

**Figure 6.** Efficiency and timing of anaphase progression are unaffected by chromosome alignment and segregation defects in human oocytes. (A) Frames from time lapse movies of live human oocytes progressing through anaphase without alignment defects (row 1) and with misaligned chromosomes (row 2 and 3). Blue arrows indicate misaligned bivalents while white arrows point to bivalents that are located at spindle poles at anaphase onset. Time represented in 00 h: 00 min format. Scale bar represents 10 μm. (B) Efficiency of progression through anaphase relative to the presence and severity of chromosome alignment defects. (C) Timing of anaphase onset relative to severity of chromosome misalignment in live human oocytes. NEBD stands for Nuclear Envelope Breakdown. (D) Timing of anaphase onset relative to incidence of lagging chromosomes in live human oocytes.

The following figure supplement is available for figure 6:

**Figure supplement 1.** Spindle remodeling in human oocytes persists after chromosome congression and decreases following chromosome alignment.

correctly attach the bivalents to the spindle. Thus, more extensive spindle reorganization may be required to correct attachment errors that arise during spindle assembly. Conversely, repeated rounds of microtubule attachment and detachment during spindle reorganization may further exacerbate splitting of sister kinetochores and cause strain on chromosome arm cohesion, promoting the twisting of bivalents and the precocious dissociation of bivalents into univalents. In this way, spindle reorganization could contribute to chromosome segregation defects in human oocytes.

Secondly, what causes the maternal age effect? We show that bivalents disintegrate precociously into univalents as women get older. Univalents were observed in more than 40% of oocytes from women over 35, suggesting that they are a major contributor to the maternal age effect, consistent with a recent study (*Sakakibara et al., 2015*). We also demonstrate that the degree of sister kinetochore separation increases with advanced maternal age. Sister kinetochore separation correlated with merotelic kinetochore-microtubule attachment. In addition, it allowed bivalents to rotate on the spindle with each of their sister-kinetochores facing towards opposite spindle poles. This is unexpected, because it is generally believed that mechanisms exist that prevent the biorientation of sister

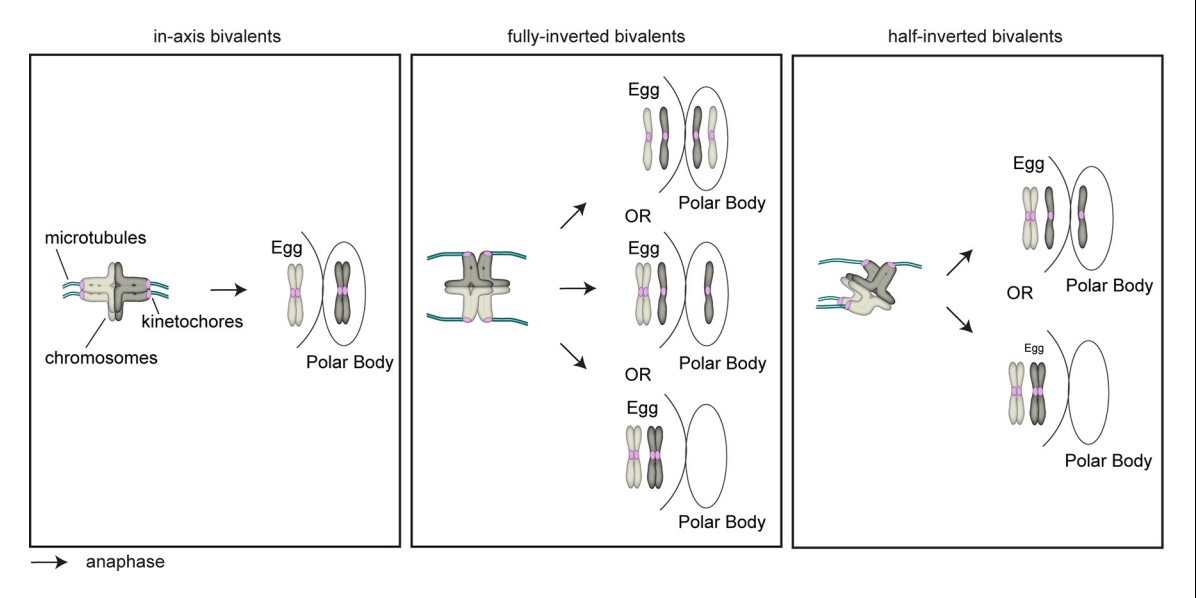

**Figure 7.** Schematic representation of bivalents with varying degree of rotation and their expected segregation outcomes following anaphase I.

kinetochores in meiosis I. We observed half- or fully-inverted bivalents in a large fraction of oocytes. It is possible that some of these abnormal attachments are corrected by the time of anaphase onset. However, it is unclear whether mechanisms exist in the oocyte that could detect these inverted bivalents and correct their attachment. Our observation that human oocytes progress into anaphase efficiently even if misaligned chromosomes are present suggests that oocytes could in principle also progress into anaphase with inverted bivalents (*Figure 7*). Such inverted bivalents could lag behind during anaphase if centromeric cohesion is still intact in one or both of the two pairs of sister kinetochores. If centromeric cohesion is compromised in one of the two sister kinetochores, the bivalent could disintegrate into one univalent and two sister chromatids. If centromeric cohesion is compromised in both pairs of sister kinetochores, the bivalent could disintegrate into four individual sister chromatids upon anaphase onset, and two non-sister chromatids of different primary parental origin

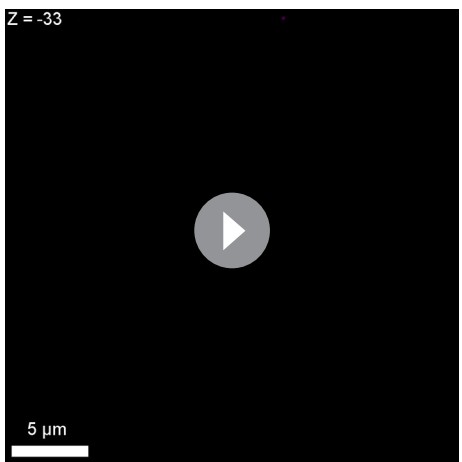

**Video 5.** A projection through confocal sections of a meiosis I spindle in an oocyte stained for Shugoshin-1 (green), chromosomes (grey), kinetochores (magenta), and in which all bivalents are oriented in axis.

would remain in the egg (*Figure 7*). Interestingly, a recent genetic study has reported that precisely this segregation outcome, which was called reverse segregation, seems to be a major cause of chromosome segregation errors during meiosis I in human oocytes (*Ottolini et al., 2015*). The mechanisms behind this reverse segregation pattern have remained unclear. Our study indicates that inverted bivalents are likely to contribute to this segregation pattern.

In addition, the precocious separation of bivalents into univalents is likely to contribute to this pattern in oocytes from both young and older women. The vast majority of univalents bioriented on the MI spindle. If the sister chromatids segregated equally to the two spindle poles upon anaphase onset, this would also result in the reverse segregation pattern that was previously observed.

Together, these data establish a mechanistic framework whereby the separated nature of

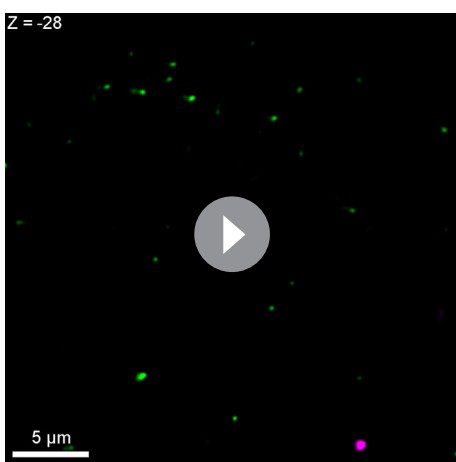

**Video 6.** A projection through confocal sections of a meiosis I spindle in an oocyte stained for Shugoshin-1 (green), chromosomes (grey), kinetochores (magenta), and in which two bivalents are fully-inverted as indicated by Shugoshin-1 staining (marked with white arrows).

sister kinetochores in human oocytes and the precocious dissociation of bivalents into univalents, combined with spindle instability (*Holubcova et al., 2015*) and unrestrained progression into anaphase, prime oocytes from women of all ages for aneuploidy.

## Materials and methods

### Preparation and culture of human oocytes

The use of immature unfertilized human oocytes in this study has been approved by the UK's National Research Ethics Service under the REC reference 11/EE/0346; IRAS Project ID 84952. Immature unfertilized oocytes were donated by women receiving assisted reproduction treatment at Bourn Hall Clinic (Cambridge, UK) between September 2012 and November 2015. 94 oocytes (from 61 donors) were used for live imaging; 87 MI oocytes (from 53 donors) and 46 MII eggs (from 29 donors) for fixations. The donors were aged between 23 and 46 years and underwent ovarian stimulation (*Faddy et al., 2011*) for intracytoplasmic sperm injection (ICSI). Couples were referred for ICSI treatment because of male factor associated infertility (69.4%), absence of a male partner (6.0%), polycystic ovary/anovulatory syndrome (5.2%), tubal damage (2.3%), endometriosis (0.7%), idiopathic infertility (11.9%) or a combination of the above/other factors (3.5%). Only oocytes that were immature at the time of the ICSI procedure and thus could not be used for in vitro fertilization were used in this study. All patients participating in this study gave informed consent for these surplus oocytes to be used in this study and for the results of this study to be published. Oocytes were collected and cultured as previously described (*Holubcova et al., 2015*). In brief, immature live oocytes were collected within 5 hr after retrieval from ovaries and transported from Bourn Hall Clinic to the MRC. The oocytes were microinjected with 10–15 μl (1–2% of human oocyte volume) of 1–2 μg/μl mRNA encoding fluorescently labelled proteins using mercury-filled needles based on previously published methods (*Jaffe and Terasaki, 2004*, *Schuh and Ellenberg, 2007*). Oocyte culture, micromanipulation and live imaging were performed in G-MOPS medium supplemented with 10% FBS under mineral oil (Paraffin, Merck) at 37°C. Only oocytes that were morphologically normal and underwent NEBD within 24 hr of retrieval from ovaries were used in this study. None of the oocytes used in this study had been freeze-thawed.

### Expression constructs and mRNA synthesis

For in vitro mRNA synthesis, pGEMHE-H2B-mRFP1 (*Schuh and Ellenberg, 2007*) and pGEMHE-EGFP-MAP4 (*Schuh and Ellenberg, 2007*) were linearized with AscI. Capped mRNA was synthesized with T7 RNA polymerase (mMessage mMachine kit, Ambion) and dissolved in 11 μl water. mRNA concentrations were determined on ethidium bromide agarose gels by comparison with an RNA standard (Ambion).

### Confocal microscopy

Time-lapse images were acquired at 37°C using a Zeiss LSM710 confocal microscope equipped with an environmental chamber and a 40x C-Apochromat 1.2 NA water immersion objective (Carl Zeiss Limited, Cambridge UK). Images were typically acquired at a temporal resolution of 5–10 min and a spatial resolution of 4 μm confocal sections covering 70 μm. We either recorded a single oocyte or multiple oocytes in parallel using Zeiss' MultiTime Series Macro. Care was taken not to expose cells to laser intensities that perturb oocyte maturation.

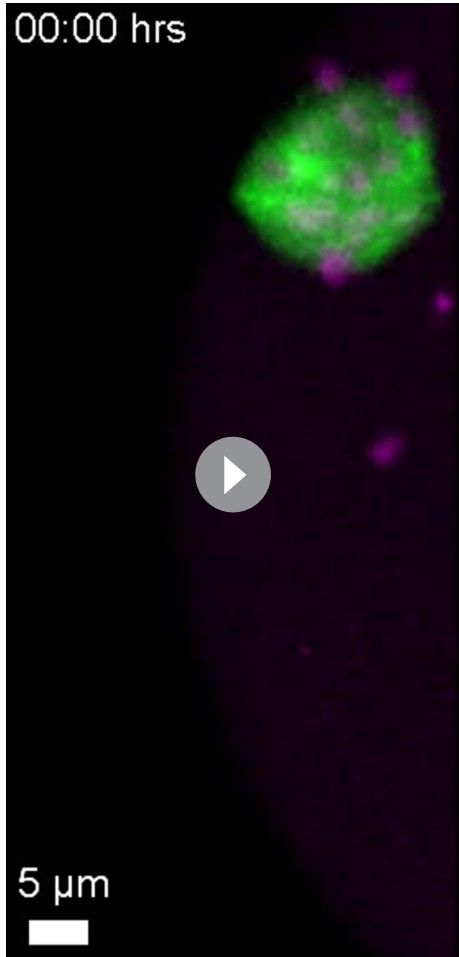

**Video 7.** Time-lapse movie of a live human oocyte progressing through anaphase I with a misaligned chromosome at the spindle pole (white arrow). Microtubules and chromosomes are labelled in green and magenta respectively.

## Cold-mediated microtubule depolymerization assays

To determine k-fiber stability, non-kinetochore-bound microtubules were depolymerized by placing oocytes at 4°C for 6 min. Cells were immediately fixed following cold treatment and processed for immunofluorescence microscopy as described below. K-fiber attachments were quantified from three-dimensional volume reconstructions of spindles using Imaris (Bitplane).

## Immunofluorescence microscopy

Oocytes were fixed for 60 min at 37°C in 100 mM HEPES (pH 7) (titrated with KOH), 50 mM EGTA (pH 7) (titrated with KOH), 10 mM MgSO$_4$, 2% formaldehyde (MeOH free) and 0.2% Triton X-100, based on previously published methods (*Strickland et al., 2004*). To analyze the morphology and orientation of bivalents close to anaphase onset, non-injected oocytes were fixed ≥15 hr after NEBD. In cases where NEBD was not evident, microinjected oocytes were assessed periodically and briefly (≤1 min) for maturation and fixed once the spindle migrated to the cell surface and acquired a size corresponding to about 15 h after NEBD, as defined by our previous study (*Holubcova et al.,*

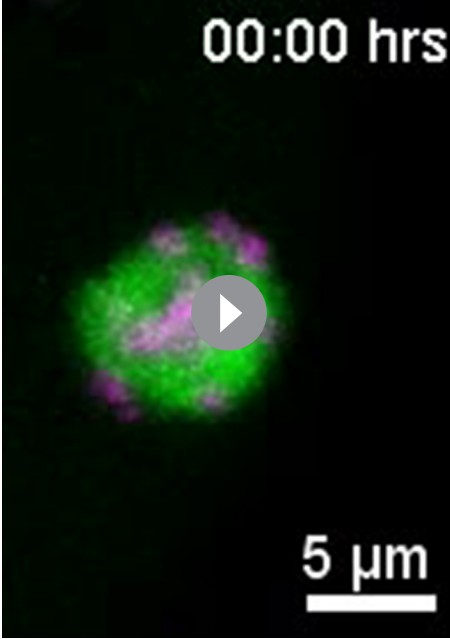

**Video 8.** Time-lapse movie of a live human oocyte progressing through anaphase I with misaligned chromosomes (white and yellow arrows). Microtubules and chromosomes are labelled in green and magenta respectively.

*2015*). None of the oocytes used for fixed cell analysis were subjected to long-term live imaging before fixation. After fixation, oocytes were extracted in PBS, 0.1% Triton X-100 overnight at 4°C. All antibody incubations were performed in PBS, 3% BSA and 0.1% Triton X-100, either overnight at 4°C (for primary antibodies) or for 3 hr at room temperature (for secondary antibodies). The primary antibodies used were rat anti-α-tubulin (MCA78G, Serotec; 1:3000), mouse anti-Hec1 (ab3613, Abcam; 1:100), mouse anti-SgoL1 (H00151648-M01, Abnova 1:100) and human ACA centromere CREST autoantibody (*FZ90C-CS1058, Europa Bioproducts;* 1:1000). As secondary antibodies, Alexa-Fluor-488/564/647 labelled anti-rat/anti-human/anti-mouse (all Molecular Probes; 1:400) were used. DNA was

stained with 0.5 µg/ml Hoechst 33342 (Molecular Probes).

Immunostained oocytes were imaged on a Zeiss LSM710 confocal microscope equipped with a 63x C Apochromat 1.2 NA water immersion objective at a spatial resolution of 0.3 µm optical sections (xy pixel 1024x1024). For analysis, images were deconvolved using Huygens Professional (Scientific Volume Imaging).

## Data analysis

### Kinetochore counting and assessment of sister pair configurations

Sister kinetochore pair configurations were determined by 3D analysis (Imaris, Bitplane) of high resolution deconvolved images encompassing the entire meiosis I spindle. First, all kinetochores belonging to the same bivalent were identified by comparing CREST and Hoechst staining in consecutive z-planes spanning the entire bivalent. Then, the two most proximal kinetochores within a bivalent were defined as a sister pair.

The center of each kinetochore was detected with subpixel accuracy using the automated spot detection function based on local maxima in Imaris (Bitplane). All automated spot detections were confirmed by visual inspection and manually corrected when software error was evident. Kinetochore configurations were defined as follows: indistinguishable—a single CREST spot visible by inspection of the maximum intensity projection together with automated single spot detection by Imaris; overlapping—a markedly stretched CREST signal in conjunction with automated detection of two spots; distinct—two discrete CREST spots that touch but do not overlap together with automated detection of two spots; separated—two discrete non-touching CREST spots identified by visual inspection together with automated detection of two spots. 98% of kinetochore pairs could be resolved with confidence and were included in the subsequent analysis.

The distances between the sister kinetochores were measured in Microsoft Excel using the xyz coordinates of kinetochore centers defined by the automated spot detection function in Imaris (Bitplane). Sister kinetochores were frequently detected in different z-sections and the Pythagorean Theorem was used to calculate the distances between all kinetochore pairs. The separation distance between all indistinguishable and overlapping sister kinetochores that were detected as single spots by Imaris was set to 0 µm (*Figure 2C*). All distances and kinetochore configurations were determined blindly to donor's age. The quantifications were further confirmed by an independent second count. Total chromosome and kinetochore counts were performed in each oocyte to exclude the possibility that any two bivalents with indistinguishable kinetochore configurations are in fact two univalents originating from a precociously dissociated bivalent. Any bivalents that were located at the spindle poles and hence were not under tension were excluded from this analysis.

### Modes of kinetochore-microtubule attachment

Microtubule-kinetochore attachments were determined by 3D analysis of high resolution deconvolved images of the whole spindle volume. Only oocytes in which the spindle was oriented in parallel to the imaging plane were included in this analysis. For quantification of K-fiber attachment modes, first the number of discrete K-fibers attaching to each sister kinetochore pair was determined. Then, if two or more K-fiber attachments were identified, the type (A-C) of the attachment was evaluated. Attachment modes were classified as: type A - two separate K-fibers that run in parallel to each other; type B - two separate K-fibers that originate from distant locations on the spindle; type C - a K-fiber that initially appears as one bundle but then branches out to form separate attachments to each of the sister kinetochores. In *Figure 2F–G*, only kinetochores with end-on attachments originating from opposite spindle poles were included in the merotelic group, whereas lateral-merotelic attachments were included in the lateral attachment group, differently from (*Holubcova et al., 2015*).

### Bivalent rotation

To determine if a bivalent was oriented in axis with the spindle (normal) or rotated relative to the spindle axis (unconventional bivalent geometry), we first determined the sister kinetochore pairs. For separated sister kinetochore pairs, the two kinetochores that were closest to each other were defined as pairs. The line that links the sister kinetochore pairs was defined as the bivalent axis.

Bivalent rotation was scored in comparison to the expected in axis orientation of a bivalent on the meiosis I spindle ('in-axis bivalent'). Bivalents in which the bivalent axis was perpendicular to the spindle axis were defined as 'fully-inverted'. In 13% of chromosomes that were scored as fully-inverted the bivalent axis could not be determined solely on the basis of the distances between the four kinetochores. Here the bivalents were also assessed for presence of kinetochore stretching (evident in other rotated bivalents) and bivalent morphology to score for bivalent rotation. In some cases, only half of the bivalent was rotated, so that one sister kinetochore pair was in axis with relation to the spindle, whereas the other sister kinetochore pair was perpendicular to the spindle axis. Such bivalents were labelled as 'half-inverted'.

### Bivalent twisting

Bivalent twisting was scored in intact bivalents based on the orientation of the sister kinetochore pairs' axes in relation to each other. By definition, the axis could only be determined in bivalents that had two sister kinetochore pairs of a split configuration. Thus, only 81% of all bivalents were scored for twisting. In a conventional undistorted bivalent, the kinetochore pairs were in parallel. A twisted bivalent configuration was a result of a rotation of one homolog within the bivalent by approximately 70 degrees in relation to the other homolog. Thus, the sister kinetochore pairs were oriented perpendicularly to one another instead of being in the same plane (i.e. in one pair, the kinetochores were position side-to-side, whereas in the other pair top-to-bottom).

### Weakened arm cohesion

Weakly attached bivalents were scored on the basis of no apparent Hoechst signal between homologous chromosomes. In oocytes with multiple bivalents showing extreme separation, the two homologs of a bivalent were identified based on their proximity to each other and to other chromosomes. Total chromosome and kinetochore counts were performed in each oocyte to confirm univalent identity. By counting the chromosomes, we could exclude the possibility that a small bivalent is scored for as a univalent. Because human oocytes in meiosis I have 46 chromosomes, to quantify presence of univalents, we first classified oocytes that presumably had more than 23 bivalents. In those oocytes, we then identified univalents based on chromosome and kinetochore morphology.

### Analysis of efficiency and timing of anaphase relative to chromosome alignment defects

To determine the stages of meiosis and assess spindle instability, chromosome segregation and alignment, time-lapse images of live human oocytes were analyzed in 3D in Imaris (Bitplane). The timing of meiotic progression was quantified relative to NEBD. Anaphase onset was defined as the time point 10 min before chromosome separation was first observed. Alignment defects were defined as follows: no- all chromosomes aligned at the time frame corresponding to anaphase onset; mild misalignment- 1–3 chromosomes misaligned at anaphase onset; severe- >3 chromosomes misaligned at anaphase onset. The degree of chromosome lagging was defined as previously described (*Holubcova et al., 2015*). The time of chromosome congression was defined as time frame when chromosomes first came together to form the metaphase plate.

### Statistics

Averages (mean) and standard deviations (s.d.) were calculated in Microsoft Excel. Unless specified otherwise, significance analyses were based on Student's *t*-test (two-tailed) and were calculated using OriginPro (OriginLab). All box plots show median (line), mean (small square), 5th, 95th (whiskers) and 25th and 75th percentiles

## Acknowledgements

We are grateful to the clinicians and nursing team at Bourn Hall Clinic who were instrumental in recruiting patients for this study and obtaining their informed consent, and also to the Embryology team for their enthusiastic support of this project from the outset and who were responsible for identifying and preparing the test oocytes.

We thank members of the Schuh lab for helpful discussions and the light microscopy facility at MRC LMB for their assistance.

The research leading to these results has received financial support from the European Research Council under grant agreement no. 337415 and from the Lister Institute of Preventive Medicine. AZ was supported by a Rosetrees Trust Fellowship. In addition, the authors were supported by the MRC.

## Additional information

### Funding

| Funder | Grant reference number | Author |
|---|---|---|
| European Research Council | 337415 | Agata P Zielinska<br>Zuzana Holubcova<br>Melina Schuh |
| Lister Institute of Preventive Medicine | Lister Prize | Agata P Zielinska<br>Zuzana Holubcova<br>Melina Schuh |
| Rosetrees Trust | PhD Fellowship | Agata P Zielinska |
| Medical Research Council | | Agata P Zielinska<br>Zuzana Holubcova<br>Melina Schuh |

The funders had no role in study design, data collection and interpretation, or the decision to submit the work for publication.

### Author contributions

APZ, generated and analysed all data with the exception of Figure 6 and Figure 6-figure supplement 1, and prepared figures; wrote the manuscript, Acquisition of data, Analysis and interpretation of data, Drafting or revising the article; ZH, generated data in Figure 6 and Figure 6-figure supplement 1 and commented on the manuscript, Acquisition of data, Analysis and interpretation of data; MB, KE, supervised the collection and handling of human oocytes in the Bourn Hall Clinic and commented on the manuscript, Drafting or revising the article, Contributed unpublished essential data or reagents; MS, prepared schematic drawings; wrote the manuscript; conceived and supervised the study, Conception and design, Analysis and interpretation of data, Drafting or revising the article

### Ethics

Human subjects: The use of immature unfertilized human oocytes in this study has been approved by the UK's National Research Ethics Service under the REC reference 11/EE/0346; IRAS Project ID 84952. Immature unfertilized oocytes were donated by women receiving assisted reproduction treatment at Bourn Hall Clinic (Cambridge, UK).

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
