## [Decision Letter]

Thank you for submitting your work entitled "Incomplete pairing of kinetochores predisposes chromosomes to segregation errors in human oocytes" for peer review at *eLife*. Your submission has been favorably evaluated by Sean Morrison (Senior editor) and three reviewers, one of whom is a member of our Board of Reviewing Editors.

The reviewers have discussed the reviews with one another and the Reviewing editor has drafted this decision to help you prepare a revised submission.

Summary:

The manuscript describes the first in-depth analysis of kinetochore behaviour during spindle assembly in human oocytes. The most surprising finding is that sister kinetochores are often separated and can form autonomous kinetochore fibres. Sister kinetochore separation in meiosis I increase with the age of the donors, suggesting that it might be a cause of mis-segregation. The authors describe interesting alignment defects that seem to correlate with increased inter-sister-kinetochore distance.

As you will see from the reviews appended below, all three referees are positive about the work, but they also raise important concerns that will need to be addressed when preparing a revised version of the manuscript.

Essential revisions:

Specifically, the referees identify three main concerns.

First, while the data provide nice correlative evidence linking increased sister kinetochore separation to chromosome mis-segregation, they also fall short of proving any causality (i.e. that increased sister separation is the effective cause of mis-segregation). The manuscript would be strongly reinforced if it provided some evidence that sister separation is indeed a significant cause of mis-segregation, in particular if live cell imaging could be used to demonstrate increased segregation error rates when sister kinetochores have undergone large separations. We realise, however, that this type of approach in human oocytes may remain technically very challenging, and therefore this addition should not be considered an absolute requirement for further consideration of the manuscript. In this case, however, the manuscript should be edited accordingly, i.e. it should clearly indicate that the study is correlative in nature.

One of the reviewers also pointed out that in your previous work you have emphasised a major role for spindle assembly problems as a cause of meiotic mis-segregation, and that the present manuscript seems to entail a considerable re-interpretation of these previous claims. A discussion of this conceptual re-alignment seems in good order.

Second, one of the reviewers advocated using at least one proper kinetochore marker (instead of the predominantly centromeric marker you have used) on a small cohort of cells to reinforce your claims of increases sister kinetochore separation.

Finally, there is a concern that your criterion to identify chromosomes with a rotated orientation on the spindle seems arbitrary, in particular because it is not clear that the shorter inter-kinetochore distance necessarily identifies the sister kinetochore pair. It would be desirable that you provided more direct evidence of this.

*Reviewer #1:*

This interesting manuscript describes how kinetochores in human oocytes influence the outcome of chromosome segregation during meiosis I. An important observation is that sister kinetochores are not fully fused in meiosis I, and that they display different degrees of separation, in a way that strongly correlates with the donor's age. Importantly, the physical separation of the sister kinetochores also leads to the establishment of separate kinetochore fibers, and in frequent kinetochore-microtubule attachment errors. The analysis required a considerable experimental tour-de-force. The study is descriptive but of great interest in that it provides first-hand insights into a fundamental cell division process in humans. The manuscript is generally well written and the results clearly described.

*Reviewer #2:*

This manuscript addresses an important question in human reproduction: the high frequency of aneuploidy in human eggs. The authors previously attributed the chromosome segregation defects to "error-prone chromosome-mediated spindle assembly" in meiosis I in their recent Science paper. The current paper argues for defects in pairing of sister kinetochores in MI as a cause of segregation errors. The same merotelic attachments that were previously interpreted as resulting from problems in spindle assembly are now interpreted as resulting from splitting of sister kinetochores. Both spindle and kinetochores could certainly contribute, but it is not clear why each paper is written from a single perspective, without discussing the other. The "spindle" paper is already published and makes a strong claim for the importance of spindle defects, so the burden is on this paper to make a convincing case that the kinetochore defects are also important.

The Abstract states, as the main result of the paper, that "human oocytes are predisposed to chromosome segregation errors because they do not pair sister kinetochores during the first meiotic division." A similar statement is made in the third sentence of the Discussion. The data do not support these claims. Sister kinetochores are farther apart in human than in mouse oocytes, but it is not justified to state that they are not paired. Furthermore (and more important), it is not clear that sisters separated by a greater distance are more likely to segregate incorrectly. Additional data would be necessary to make this claim convincing, such as live imaging showing that "split" kinetochores are more likely to end up as lagging chromosomes in anaphase or to segregate incorrectly.

*Reviewer #3:*

Compared to mitosis, chromosome segregation during meiosis in human oocytes is surprisingly error prone. However, the mechanistic basis for these defects is unclear. Previous work has implicated defects in sister chromatid cohesion as accounting for at least some of the age-dependent increases in meiotic chromosome segregation errors. In this paper, Zielinska et al. additionally implicate kinetochore pairing as a source of meiotic errors by showing that many sister kinetochores in meiosis I oocytes of young donors are split into two distinct units rather than unified as in mouse oocytes. The majority of these split sister kinetochores are attached to distinct kinetochore fibers, indicating that the spindle recognizes them as separate kinetochores. Intriguingly, the proportion of separated kinetochores found in oocytes from older donors was higher than those from younger donors. Additionally, separated kinetochores are more likely to form improper merotelic attachments with associated kinetochore fibers. This work also found that bivalents where one or more of the sister kinetochore pairs are separated are able to rotate and twist on the spindle. Many bivalents also had gaps between chromosome pairs or had even separated completely into univalents, indicating a reduction in cohesion along chromosome arms. Univalents were found more often in oocytes from older donors. The authors found that a large fraction of oocytes had misaligned chromosomes at anaphase onset, but the vast majority of these did not undergo a metaphase I arrest, but instead progressed into anaphase without significant delay.

This work provides exciting new insights into the arrangement of sister kinetochores in human oocytes, as well as the status of their associated bivalents. This work also provides possible explanations behind both the error-prone nature of human female meiosis and the maternal age effect. However, although the observations made in this paper are of high quality, the causal relationships between the various data points remain elusive. Ultimately, the data in this paper are primarily observational, although this is intimately related to the nature of the system that they are using (oocytes from human donors). To my knowledge, no one else is doing this type and high quality of work in human oocytes, and the broad importance of this work clearly justifies its publication. Nonetheless, I believe that some carefully selected changes to the paper and a few additional analyses would significantly improve the impact of this work.

1) For this paper, the authors essentially conduct two types of analyses: fixed immunofluorescence imaging of oocytes and live cell analysis of meiotic progression. In each case, they visualize these using a limited selection of markers. I feel that the histone and tubulin markers used for the live cell imaging are sufficient to visualize what they want to see in this case, and nicely justify their main points related to meiotic progression. However, as a major focus of this work is the nature of the connections between the two sister kinetochores on a homologue and their attachment to the spindle, I am not sure that CREST staining provides the best marker for kinetochores in the fixed cell case. CREST stains a variety of centromere and kinetochore proteins, and as it is derived from different human autoimmune patients it is never exactly clear what a batch corresponds to. The primary protein that is likely to be detected here is CENP-B, which is not the best marker for kinetochore function. If possible, I would strongly suggest that the authors repeat this analysis for a very small and limited set of oocytes (I appreciate the complexity of these experiments and how precious these oocytes are) to confirm these results using an antibody targeted against a constitutive centromere component, such as CENP-A or CENP-C, and an outer kinetochore component, such as Ndc80. Excellent commercial antibodies exist for these, and it would help increase the impact of their findings. For example, it is technically possible that one of these kinetochores is inactivated, at least in some cases. Such a kinetochore would stain for CENP-B, but not for Ndc80, for example.

2) A key feature that is missing from this paper is the ability to conduct a functional analysis to test the consequences of their observations. Fortunately, there is a beautiful recent Nature Genetics paper by Ottolini et al. (2105), which the authors nicely and appropriately cite, which describes the products of meiosis and reveals a surprisingly common pattern of reverse meiosis (with the sister chromatids segregating first). This is consistent with the results presented here. However, in addition to frequent sister separation in MI, this requires this behavior to be compensated for in MII. In this case, the sister separation would not be a cause of errors, but would lead instead to a different meiotic program. It would be interesting to consider (and visualize if possible, although again I understand the limitations here) what happens in MII for the cells that precociously separate sister chromatids in MI.

3) Related to point 2, I would suggest that the authors at least slightly reword their paper as they have not definitively demonstrated that this sister kinetochore separation is the source of aneuploidy and segregation errors in human meiosis. As it stands, the Abstract and title, for example, are worded to imply a causal relationship. I would urge caution such that the authors propose these ideas, but use wording that suggests these, instead of demonstrating it. Similarly, although the authors cite prior work suggesting that a reduction in sister chromatid cohesion contributes to the age dependent increase in maternal aneuploidy, they don't specifically address this idea in the text. Based on the potential mechanisms for sister kinetochore pairing in MI, there is the possibility of an intimate linkage between cohesion defects and kinetochore orientation problems.

4) The authors do not address whether the increase in merotelic attachments with age shown in Figure 2 is completely attributable to the increase in split kinetochores. There may be additional factors that cause either unified or distinct kinetochores in older donors to more often form merotelic attachments. An additional analysis that breaks down the kinetochores with merotelic attachments by both age and kinetochore configuration would address this. In this case, I would hope that the authors would be able to present this data based on the oocytes that they have already imaged and without conducting additional experiments.

5) In the Results, the authors suggest that separated sister kinetochores increase the risk of merotelic kinetochore attachments. However, the causal relationship is not clear from the data presented. It is possible that merotelic attachments pull closely configured sister kinetochores farther apart. Sister kinetochores should be observed at earlier time points, after NEBD but before a bipolar spindle has fully formed, to determine if separated sister kinetochores appear under conditions of little bidirectional pulling force.

6) The proportion of chromosome pairs with gaps between the bivalents that are rotated or twisted should be presented. If the proportion were high, this would add merit to the idea that twisting forces put strain on cohesion at chromosome arms.

7) It is unclear how frequently oocytes entered anaphase with misaligned chromosomes in the live imaging experiments. A figure detailing the percent of oocytes found with mild or severe chromosome misalignments should be shown. Additionally, if this data were broken up by donor age, potentially interesting insights into the maternal age effect could be found.

---

## [Author Response]

*Reviewer #2:*

*This manuscript addresses an important question in human reproduction: the high frequency of aneuploidy in human eggs. The authors previously attributed the chromosome segregation defects to "error-prone chromosome-mediated spindle assembly" in meiosis I in their recent Science paper. The current paper argues for defects in pairing of sister kinetochores in MI as a cause of segregation errors. The same merotelic attachments that were previously interpreted as resulting from problems in spindle assembly are now interpreted as resulting from splitting of sister kinetochores. Both spindle and kinetochores could certainly contribute, but it is not clear why each paper is written from a single perspective, without discussing the other. The "spindle" paper is already published and makes a strong claim for the importance of spindle defects, so the burden is on this paper to make a convincing case that the kinetochore defects are also important.*

We thank the reviewer for highlighting this important point. We had initially drafted this manuscript in a different format, where length restrictions precluded more extensive discussions on how spindle instability and the split nature of sister kinetochores in human oocytes could be linked. In the revised manuscript, we have substantially extended the Introduction and Discussion to discuss this point. We also explain how the interplay between spindle instability and kinetochore separation may promote merotelic kinetochore- microtubule attachments in the following paragraph in the Discussion:

“Our previous work established that the spindles in human oocytes form in a lengthy and error-prone process, during which the spindle undergoes extensive reorganization. […] In this way, spindle reorganization could contribute to chromosome segregation defects in human oocytes.”

*The Abstract states, as the main result of the paper, that "human oocytes are predisposed to chromosome segregation errors because they do not pair sister kinetochores during the first meiotic division." A similar statement is made in the third sentence of the Discussion. The data do not support these claims. Sister kinetochores are farther apart in human than in mouse oocytes, but it is not justified to state that they are not paired.*

We thank the referee for highlighting that we need to be more cautious with the terminology that we use to refer to split sister kinetochores. We have modified the sections mentioned above as follows:

“Here, we show that human oocytes are predisposed to chromosome segregation errors because they do not pair sister kinetochores during the first meiotic division.”

Now reads:

“Here, we show that many sister kinetochores in human oocytes are separated and do not behave as a single functional unit during the first meiotic division.”

“Our findings establish that the unpaired nature of sister kinetochores biases oocytes from women of all ages towards chromosome segregation errors.”

Now reads:

“The data presented in this study suggest that merotelic attachments are also promoted by the split nature of sister kinetochores in human oocytes.”

*Furthermore (and more important), it is not clear that sisters separated by a greater distance are more likely to segregate incorrectly. Additional data would be necessary to make this claim convincing, such as live imaging showing that "split" kinetochores are more likely to end up as lagging chromosomes in anaphase or to segregate incorrectly.*

We thank the reviewer for this very insightful comment. Unfortunately, we cannot currently image kinetochores in human oocytes by live cell microscopy. Our observation that the degree of sister kinetochore separation correlates with the probability of merotelic attachments suggests that the separation of sister kinetochores should increase the chances of chromosome segregation errors. We agree though that further data from live oocytes are required to show that this correlation is of a causal nature and have thus toned down this point in the title, Abstract and the main text, highlighting instead that our findings are of a correlative nature.

*Reviewer #3:*

*1) For this paper, the authors essentially conduct two types of analyses: fixed immunofluorescence imaging of oocytes and live cell analysis of meiotic progression. In each case, they visualize these using a limited selection of markers. I feel that the histone and tubulin markers used for the live cell imaging are sufficient to visualize what they want to see in this case, and nicely justify their main points related to meiotic progression. However, as a major focus of this work is the nature of the connections between the two sister kinetochores on a homologue and their attachment to the spindle, I am not sure that CREST staining provides the best marker for kinetochores in the fixed cell case. CREST stains a variety of centromere and kinetochore proteins, and as it is derived from different human autoimmune patients it is never exactly clear what a batch corresponds to. The primary protein that is likely to be detected here is CENP-B, which is not the best marker for kinetochore function. If possible, I would strongly suggest that the authors repeat this analysis for a very small and limited set of oocytes (I appreciate the complexity of these experiments and how precious these oocytes are) to confirm these results using an antibody targeted against a constitutive centromere component, such as CENP-A or CENP-C, and an outer kinetochore component, such as Ndc80. Excellent commercial antibodies exist for these, and it would help increase the impact of their findings. For example, it is technically possible that one of these kinetochores is inactivated, at least in some cases. Such a kinetochore would stain for CENP-B, but not for Ndc80, for example.*

We thank the reviewer for this very useful remark. In order to investigate whether the separation is also visible in the outer-kinetochore region that interacts with microtubules, we performed analysis of Hec1 localization in five additional human oocytes (Figure 1—figure supplement 1). The co-staining of Hec1 together with CREST shows that the separation of kinetochores is also true for the outer kinetochore region, as shown by the localization of this Ndc80 component. We show that the pattern of staining for CREST and Hec1 is very similar. Moreover, we observed all types of kinetochore configurations with Hec1 staining which we have previously observed with CREST labelling (Figure 1). Due to the scarce supply of samples and the difficulty in finding antibodies that work in oocytes, we were unfortunately not able to find an inner kinetochore antibody that works reliably in human oocytes. Nonetheless, investigating the distribution of the inner versus outer kinetochore markers in human oocytes is of high importance and is a very interesting point to address in future studies.

*2) A key feature that is missing from this paper is the ability to conduct a functional analysis to test the consequences of their observations. Fortunately, there is a beautiful recent Nature Genetics paper by Ottolini et al. (2105), which the authors nicely and appropriately cite, which describes the products of meiosis and reveals a surprisingly common pattern of reverse meiosis (with the sister chromatids segregating first). This is consistent with the results presented here. However, in addition to frequent sister separation in MI, this requires this behavior to be compensated for in MII. In this case, the sister separation would not be a cause of errors, but would lead instead to a different meiotic program. It would be interesting to consider (and visualize if possible, although again I understand the limitations here) what happens in MII for the cells that precociously separate sister chromatids in MI.*

We agree that it would be of great interest to image chromosomes live as they are segregating and to observe how this reverse segregation pattern arises, and which mechanisms may potentially be involved in compensating for it during the second meiotic division. Unfortunately though, we cannot currently image kinetochores and chromosomes with sufficient resolution to comment on this problem. We hope that we will be able to address this point in future studies as it is indeed a very interesting point.

*3) Related to point 2, I would suggest that the authors at least slightly reword their paper as they have not definitively demonstrated that this sister kinetochore separation is the source of aneuploidy and segregation errors in human meiosis. As it stands, the Abstract and title, for example, are worded to imply a causal relationship. I would urge caution such that the authors propose these ideas, but use wording that suggests these, instead of demonstrating it. Similarly, although the authors cite prior work suggesting that a reduction in sister chromatid cohesion contributes to the age dependent increase in maternal aneuploidy, they don't specifically address this idea in the text. Based on the potential mechanisms for sister kinetochore pairing in MI, there is the possibility of an intimate linkage between cohesion defects and kinetochore orientation problems.*

We thank the reviewer for this very insightful comment. Unfortunately, we cannot currently image kinetochores in human oocytes by live cell microscopy. Our observation that the degree of sister kinetochore separation correlates with the probability of merotelic attachments suggests that the separation of sister kinetochores should increase the chances of chromosome segregation errors. We agree though that further data from live oocytes are required to show that this correlation is of a causal nature and have thus toned down this point in the title, Abstract and the main text, highlighting instead that our findings are of a correlative nature.

We have now also included the possible involvement of cohesion in the defects we observe. The corresponding section now reads:

“Recent work in mouse oocytes suggests that the cohesin complex is gradually lost from chromosomes in oocytes as mice get older (Lister et al., 2010, Chiang et al., 2010. […] Whether such a loss of cohesin is also relevant in human oocytes is still unclear (Garcia-Cruz et al., 2010, Tsutsumi et al., 2014)”.

*4) The authors do not address whether the increase in merotelic attachments with age shown in Figure 2 is completely attributable to the increase in split kinetochores. There may be additional factors that cause either unified or distinct kinetochores in older donors to more often form merotelic attachments. An additional analysis that breaks down the kinetochores with merotelic attachments by both age and kinetochore configuration would address this. In this case, I would hope that the authors would be able to present this data based on the oocytes that they have already imaged and without conducting additional experiments.*

We thank the reviewer for this excellent suggestion. We have now presented the fraction of merotelic attachments, based both on the kinetochore configuration and donor’s age (Figure 2—figure supplement 3). We see that with an increase in separation (unified versus distinct versus separated), there is an increase in merotely regardless of age. However, this analysis also revealed that separated kinetochores may be more likely to be merotelically attached in oocytes from older donors than in oocytes from younger donors. It is possible that the degree of separation affects the likelihood of merotelic attachments. But also additional age-related changes in oocytes may promote the observed increase, as rightly highlighted by the reviewer. We now discuss both of these possibilities in the revised version of the manuscript:

“The number of merotelic kinetochore-microtubule attachments also increased with maternal age, from around 7% in women under 30 to around 21% in women over 35 (Figure 2). […] This could be explained by the increase in the degree of separation between sister kinetochores as women get older (Figure 2), as well as additional age related defects that could contribute to aberrant attachments.”

*5) In the Results, the authors suggest that separated sister kinetochores increase the risk of merotelic kinetochore attachments. However, the causal relationship is not clear from the data presented. It is possible that merotelic attachments pull closely configured sister kinetochores farther apart. Sister kinetochores should be observed at earlier time points, after NEBD but before a bipolar spindle has fully formed, to determine if separated sister kinetochores appear under conditions of little bidirectional pulling force.*

We thank the referee for raising this interesting point. At this stage it is difficult to unequivocally state the degree to which spindle forces contribute to sister kinetochore separation. This question would ideally be addressed by high resolution live imaging of kinetochores and scoring the separation in the same oocyte at different time points. This is however experimentally very challenging and could not yet be achieved in our lab for human oocytes. Moreover, because human oocytes nucleate microtubules from chromosomes (Holubcova et al., 2015), the morphology of human bivalents together with the compactness of chromosomes during the initial stages of spindle remodeling make it practically impossible to reliably measure all the distances at the early stages. Also, because the kinetochores may already be separated at this stage, the high proximity of the bivalents before the bipolar spindle assembles could lead to an incorrect assignment of kinetochores to the bivalents. Because human bivalents vary greatly in size, we also believe that it would not be representative to measure the inter kinetochore distances only in a fraction of bivalents that are on the surface of the remodeling chromatin mass and are hence more accessible to investigations.

We have however now fixed oocytes at early time points after NEBD, and what we can state with certainty is that even at those early time points sister kinetochores in human oocytes challenge the dogma that kinetochores are fused. We frequently observed kinetochores of the distinct configuration (Figure 1—figure supplement 2). It is well possible though that the separation increases even further as the oocytes progress through meiosis I. This is definitely a very interesting point to be investigated in the future. We also extended the potential interplay between spindle remodeling and kinetochore separation in the Discussion section, so that it now reads:

“It is conceivable that the specialized architecture of bivalents in human oocytes and spindle reorganization are directly linked: the fact that the four sister kinetochores of a bivalent chromosome can interact with microtubules independently may make it more challenging for the oocyte to correctly attach the bivalent to the spindle. […] Conversely, repeated rounds of microtubule attachment and detachment during spindle reorganization may further exacerbate splitting of sister kinetochores and cause strain on chromosome arm cohesion, promoting the twisting of bivalents and the precocious dissociation of bivalents into univalents.”

Moreover, we have now addressed in the text the possibility that other factors than just the kinetochore configuration may promote merotelic attachments:

“Separated kinetochores were more likely to be merotelically attached in oocytes from older women than in oocytes from young women (Figure 2—figure supplement 3). This could be explained by the increase in the degree of separation between sister kinetochores as women get older (Figure 2), as well as additional age related defects that could contribute to aberrant attachments.”

*6) The proportion of chromosome pairs with gaps between the bivalents that are rotated or twisted should be presented. If the proportion were high, this would add merit to the idea that twisting forces put strain on cohesion at chromosome arms.*

This is an excellent suggestion. Following the reviewer’s suggestion, we have now performed additional analysis on the correlation between the twisting and gaps in the Hoechst signal. Even though bivalents with gaps were more likely to be twisted than intact bivalents, the difference was not significant. This may be due to a relatively small number of bivalents with gaps that we have observed (n=84). Moreover, bivalents exhibited different degrees of separation, and pooling all bivalents with gaps into the same category may mask a potential difference in bivalents with larger gaps. We have now edited the text of the manuscript to tone down the statement and also provided a more extensive discussion on how the deterioration of the cohesion complex that has been observed in aged mice may explain our observations.

“Such twisting is likely to exert forces on arm cohesion, and may promote the precocious dissociation of bivalents into individual chromosomes, frequently referred to as univalents in meiosis (Tachibana-Konwalski et al., 2010). Consistent with this idea, arm cohesion was reduced in a large fraction of chromosome bivalents (Figure 4).”

Now reads:

“Such twisting is likely to exert additional forces on arm cohesion, which is possibly already weakened in oocytes at the time of meiotic resumption (Duncan et al., 2012, Chiang et al., 2010a). Arm cohesion was indeed reduced in a large fraction of chromosome bivalents (Figure 5).”

*7) It is unclear how frequently oocytes entered anaphase with misaligned chromosomes in the live imaging experiments. A figure detailing the percent of oocytes found with mild or severe chromosome misalignments should be shown. Additionally, if this data were broken up by donor age, potentially interesting insights into the maternal age effect could be found.*

We thank the reviewer for raising this point. The fraction of oocytes that progress through anaphase with misaligned chromosomes is presented in Figure 6. As the graph did not show the contribution of oocytes with different degrees of alignment defects to the total number of oocytes that we have analyzed, we have now revised this figure to include the n numbers for each chromosome alignment category. We agree that breaking up the data based on the donors’ age would have been an excellent idea. However, the numbers of oocytes from young and older donors in each category was very low so that better statistics are needed to draw meaningful conclusions. We agree that it would have been nice to also discuss the results in Figure 6 in terms of age as the rest of the manuscript focusses on how age affects bivalent and kinetochore morphology. But unfortunately, we do not currently have sufficient data to address this point. An alternative would be to omit Figure 6 from the current manuscript and to discuss the data in a more extensive study in the future instead.